# Single-cell RNA-seq of heart reveals intercellular communication drivers of myocardial fibrosis in diabetic cardiomyopathy

Wei Li[1], Xinqi Lou[2], Yingjie Zha[2], Yinyin Qin[2], Jun Zha[3], Lei Hong[2], Zhanli Xie[2], Shudi Yang[4], Chen Wang[3], Jianzhong An[2], Zhenhao Zhang[2]*, Shigang Qiao[2,3]*

[1]Cyrus Tang Hematology Center, Soochow University, Suzhou, China; [2]Institute of Clinical Medicine Research, Suzhou Science & Technology Town Hospital, Gusu School, Nanjing Medical University, Suzhou, China; [3]Faculty of Anesthesiology, Suzhou Science & Technology Town Hospital, Gusu School, Nanjing Medical University, Suzhou, China; [4]Suzhou Polytechnic Institute of Agriculture, Suzhou, China

*For correspondence:
zhangzhh1@njmu.edu.cn (ZZ);
qiaos@njmu.edu.cn (SQ)

Competing interest: The authors declare that no competing interests exist.

**Abstract** Myocardial fibrosis is the characteristic pathology of diabetes-induced cardiomyopathy. Therefore, an in-depth study of cardiac heterogeneity and cell-to-cell interactions can help elucidate the pathogenesis of diabetic myocardial fibrosis and identify treatment targets for the treatment of this disease. In this study, we investigated intercellular communication drivers of myocardial fibrosis in mouse heart with high-fat-diet/streptozotocin-induced diabetes at single-cell resolution. Intercellular and protein–protein interaction networks of fibroblasts and macrophages, endothelial cells, as well as fibroblasts and epicardial cells revealed critical changes in ligand–receptor interactions such as Pdgf(s)–Pdgfra and Efemp1–Egfr, which promote the development of a profibrotic microenvironment during the progression of and confirmed that the specific inhibition of the Pdgfra axis could significantly improve diabetic myocardial fibrosis. We also identified phenotypically distinct *Hrc*hi and *Postn*hi fibroblast subpopulations associated with pathological extracellular matrix remodeling, of which the *Hrc*hi fibroblasts were found to be the most profibrogenic under diabetic conditions. Finally, we validated the role of the *Itgb1* hub gene-mediated intercellular communication drivers of diabetic myocardial fibrosis in *Hrc*hi fibroblasts, and confirmed the results through AAV9-mediated *Itgb1* knockdown in the heart of diabetic mice. In summary, cardiac cell mapping provides novel insights into intercellular communication drivers involved in pathological extracellular matrix remodeling during diabetic myocardial fibrosis.

## Editor's evaluation

In this work, utilizing a murine model as well as in vitro studies, the authors provide novel insights about drivers of intercellular communication underlying pathological extracellular matrix remodeling during diabetic cardiomyopathy. The results of the study will be of interest to investigators interested in diabetic cardiomyopathy and heart failure as well as those looking for new potential targets in the treatment of heart failure.

## Introduction

Diabetes is the third most common threat to human health, approximately 537 million adults living with diabetes, of which type 2 diabetes patients account for more than 90% of all diabetic patients.

Cardiac complications are the most common causes of death and disability associated with diabetes. As the key initiating factor of diabetic cardiomyopathy, hyperglycemia can prevent optimal utilization of glucose by cardiomyocytes, which leads to myocardial fibrosis. Myocardial fibrosis is characterized by the increase in extracellular matrix proteins, deposition of interstitial collagen, disarrangement of cardiomyocytes, and the remodeling of cardiac structure (*Russo and Frangogiannis, 2016*; *Jia et al., 2018*). Since adult mammalian cardiomyocytes are incapable of regeneration, the most extensive pathological extracellular matrix remodeling and fibrosis occur during various heart diseases caused by cardiomyocyte death (*Kong et al., 2014*). Understanding the mechanisms responsible for myocardial fibrosis is critical to develop anti-fibrotic therapy strategies for diabetic patients.

Mammalian hearts consist of many cell types, including cardiomyocytes, macrophages, fibroblasts and endothelial cells, etc. (*Banerjee et al., 2007*; *Litviňuková et al., 2020*). Cell-to-cell communication is a fundamental feature of adult complex organs. These different types of cells communicate through interactions with ligand–receptor, where a ligand may be secreted and bind to a receptor, or through the fusion of two adjacent interacting cell membranes (*Ramilowski et al., 2015*). The maintenance of heart homeostasis depends on intercellular communication (*Ramilowski et al., 2015*). Many ligand–receptor signaling patterns have been identified between cardiac cells, indicating the critical role of intercellular communication in many pathophysiological processes. Therefore, intercellular communication has become a powerful therapeutic target for prevention or reversion of some of the damaging consequences of diabetic myocardial fibrosis by maintaining fine-tuned intercellular communication among different cardiac cells. Despite these findings, the overall effect of diabetes on cardiac intercellular communication and myocardial fibrosis remains poorly understood.

Single-cell RNA sequencing (scRNA-seq) is a feasible strategy for studying the cellular heterogeneity of any organ since it allows transcriptomic profiling of individual cells (*Butler et al., 2018*; *Gladka et al., 2018*; *Litviňuková et al., 2020*; *McLellan et al., 2020*). Recent scRNA-seq of many tissues has revealed cellular heterogeneity and novel intercellular crosstalk among different cell types. In this study, we developed a diabetic mouse model through high-fat diet (HFD) combined with streptozotocin (STZ) administration, and identified the composition of all cardiac cells, cellular subpopulations associated with diabetic cardiomyopathy, enrichment of signaling pathways involved in myocardial fibrosis, alterations of key ligand–receptor interactions and the most profibrotic *Hrc*[hi] fibroblast subcluster in mouse hearts using scRNA-seq on a 10× genomics platform. Our study suggests that cardiac intercellular communication plays a critical role in diabetic myocardial fibrosis and specific targeting of *Hrc*[hi] fibroblasts may be a potential therapeutic target for this cardiac disease.

## Results
### Single-cell profile of heart in diabetic mice

Conventional single-cell RNA-seq does not cover all cells in the rodent myocardium for subsequent deep sequencing (*Skelly et al., 2018*; *Forte et al., 2020*). Therefore, we isolated the nuclear fractions of all cardiac cells to assess the heterogeneity of cell populations and changes in the transcriptional profile in response to the pathology of HFD/STZ-induced diabetes (*Figure 1—figure supplement 1A*; *Grindberg et al., 2013*; *Ackers-Johnson et al., 2018*; *Lake et al., 2019*; *Koenig et al., 2022*). We classified 32,585 cardiac cells from 6 healthy controls (16,490 cells) and 6 HFD/STZ-induced diabetic mice (16,095 cells) into 25 transcriptionally distinct pre-clusters that exhibited highly consistent expression patterns across individual mice (*Figure 1—figure supplement 1B* and *Supplementary file 1*), and identified 14 populations (*Figure 1A*) based on cell-specific markers and significantly enriched genes. The cell populations included fibroblasts (*Pdgfra*, *Pcdh9*, *Bmper*), endothelial cells (*Pecam1*, *Ccdc85a*, *Btnl9*), macrophages (*Fcgr1*, *F13a1*, *Adgre1*), pericytes (*Pdgfrb*, *Vtn*, *Trpc3*), adipocytes (*Adipoq*, *Plin1*, *Tshr*), cardiomyocytes (*Ttn*, *Mhrt*, *Myh6*), smooth muscle cells (*Acta2*, *Myh11*, *Cdh6*), endocardial cells (*Npr3*, *Tmem108*, *Plvap*), epicardial cells (*Msln*, *Pcdh15*, *Muc16*), schwan cells (*Plp1*, *Gfra3*), and other immune cell populations (T cells, monocytes, B cells) (*Figure 1B*). Based on cell types, markers, and relative proportions, we concluded that our data were robust and consistent with previous 10× single nucleus data from mice heart (*McLellan et al., 2020*; *Koenig et al., 2022*). Further examination of established marker genes in each cardiac cell population revealed the presence of a wide range of cell types in all analyzed hearts (*Figure 1C, D*).

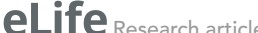

**Figure 1.** Single-cell profile of heart in diabetic mice. (**A**) t-SNE projection of all mouse cardiac cells (*n* = 16,490 cardiac cells from 6 control mice and *n* = 16,095 cardiac cells from 6 diabetic mice). (**B**) The marker genes defining each type of cell cluster in A are listed. (**C**) Heatmap showing the canonical cell markers of major cardiac cell populations. (**D**) Dot plot representing the top 5 distinct genes in each cell population. (**E**) Lollipop plot showing the number of up- and downregulated genes in high-fat diet (HFD)/streptozotocin (STZ)-treated mouse heart cells, compared with the control. Adipo, adipocytes; Cardio, cardiomyocytes; Endocar, endocardial cells; EC, endothelial cells; Epicar, epicardial cells; Fibro, fibroblasts; LEC, lymphatic ECs; Mono, monocytes; Schwann, schwann cells; SMC, smooth muscle cells; Macro, macrophages; Peri, pericytes; Details of the 25 transcriptionally distinct pre-clusters with highly consistent expression patterns across individual mouse heart are listed in *Supplementary file 1*. Detailed genes with significant transcriptomic changes in cardiac populations are listed in *Supplementary file 2*. The details of top 10 upregulated genes in cardiac populations are listed in *Supplementary file 3*.

The online version of this article includes the following figure supplement(s) for figure 1:

**Figure supplement 1.** Experimental design and cell type characterization.

**Figure supplement 2.** The top 10 upregulated genes during the pathology of high-fat diet (HFD)/streptozotocin (STZ)-induced diabetes within each cell population.

**Figure supplement 3.** Top 30 enriched Kyoto Encyclopedia of Genes and Genomes (KEGG) pathways in high-fat diet (HFD)/streptozotocin (STZ)-treated mouse fibroblasts.

HFD/STZ treatment induced significant transcriptomic changes in most cardiac populations, especially in fibroblasts, endothelial cells, endocardial cells, cardiomyocytes and macrophages (*Figure 1E*, *Supplementary file 2 and* two-sided Wilcoxon rank-sum test, FDR ≤0.05, log2FC ≥0.36). Analysis of the top 10 upregulated genes during the pathology of HFD/STZ-induced diabetes within each cell population showed that some of the top 10 upregulated genes show a noticeable increase in

expression across many different cell types, even though some are primarily expressed in only one cell type (*Figure 1—figure supplement 2* and *Supplementary file 3*). Among the top upregulated genes in response to HFD/STZ-induced diabetes within each cell population were transcripts for *Pdk4*, *Angptl4*, *Txnip*, *Postn*, *Hmgcs2*, and *Ucp3*, of which several have been previously involved in heart failure (*Yoshioka et al., 2007*; *Lang et al., 2018*; *Sheeran et al., 2019*; *Koenig et al., 2022*). The top 30 KEGG (Kyoto Encyclopedia of Genes and Genomes) pathways within fibroblast population were associated with dilated cardiomyopathy, cardiac muscle contraction, and hypertrophic cardiomyopathy (*Figure 1—figure supplement 3*, two-sided Wilcoxon rank-sum test, FDR ≤0.05). These results indicate a significantly increased risk of cardiovascular diseases in a diabetic setting.

Taken together, scRNA-seq identified distinct cell populations in the mouse heart that can help characterize HFD/STZ-induced diabetes-related changes based on gene expression and quantify gene–trait associations.

## Effects of HDF/STZ-induced diabetes on cardiac intercellular communication

Cell subpopulations with non-overlapping functions present distinct transcriptomic perturbations in response to pathological stimuli (*Mathys et al., 2019*). To determine whether distinct cardiac cell populations respond heterogeneously to diabetic stimuli, we compared differentially expressed genes in all cardiac cell types and identified 2118 unique differentially expressed genes (uni-DEGs) associated with each of the major cardiac cell types (*Supplementary file 4*). Most of the uni-DEGs (96.6%) were detected in cardiomyocytes (32.8%), fibroblasts (18.5%), macrophages (17.7%), endothelial cells (19.7%), and endocardial cells (7.9%). The genes that were most highly expressed in only a single-cell type were *Gm20658* (fibroblasts), *Ucp3* (cardiomyocytes), *Spock2* (endocardial cells), *Irf7* (endothelial cells), and *Ifi206* (macrophages), of which *Ucp3* and *Irf7* were involved in heart failure and pathological cardiac hypertrophy (*Jiang et al., 2014*; *Senatus et al., 2020*). However, the association of *Gm20658*, *Spock2* and *Ifi206* with myocardial fibrosis or heart failure has not been previously reported. Gene Ontology (GO) analysis of the uni-DEGs (upregulated) showed that terms associated with collagen fibril organization and extracellular matrix remodeling were enriched in the cardiac fibroblasts (*Figure 2A* , two-sided Wilcoxon rank-sum test, FDR ≤0.05), indicating that fibroblasts are key cellular contributors to extracellular matrix remodeling and cardiac fibrosis.

The proper functioning of metazoans is tightly controlled by intercellular communication between multiple cell populations, which is based on interactions between secretory ligands and receptors (*Ramilowski et al., 2015*). To determine the effect of HFD/STZ-induced diabetes on cardiac intracellular communication, we mapped the intercellular connection network of the cardiac cellulome in healthy controls and diabetic mice. Initially, we identified genes that were differentially expressed in specific cell populations in the mouse heart, focusing on those that were overexpressed in a single-cell type, that is, specific highly expressed genes, at FDR ≤0.01 with a minimum twofold difference in expression (*Supplementary file 5*). The gene expression patterns for receptors and ligands were found to be cell type specific in the heart secretome genes analyzed through clustering (*Figure 2—figure supplement 1A, B*, *Supplementary file 6*, *Supplementary file 7*). Analysis of the factors that support the growth of specific cell populations revealed critical intercellular communication channels. These include signaling pathways that support the survival of specific cell populations of the mouse hearts (*Figure 2B*, *Supplementary file 8*). For example, pericytes and fibroblasts express *Il34* and *Csf1*, respectively (*Figure 2B*), and communicate through *Csf1r*, while functioning as key factors for macrophage survival and growth. Fibroblasts also express *Igf1* and *Ngf* (*Figure 2B*), which support the growth of endothelial cells, mural cells, and neurons (*Glebova and Ginty, 2004*; *Bach, 2015*). To construct a map of intercellular signaling among heart cells using scRNA-seq data, we integrated them with a ligand–receptor interaction database (*Ramilowski et al., 2015*). The results showed that the endothelial and fibroblast clusters are prominent hubs for autocrine and paracrine signaling (*Figure 2C, D*, *Supplementary file 9*, *Supplementary file 10*, FDR ≤0.01, log2FC ≥1), and that intercellular signaling in response to HFD/STZ-induced diabetes had changed in all cardiac cells, with fibroblasts showing the greatest increase in the number of connections (*Figure 2E–G*, *Supplementary file 11*, *Supplementary file 12* and two-sided Wilcoxon rank-sum test, FDR ≤0.05, log2FC ≥0.36). Together, the results of these analyses suggest that intercellular communications play an important role in the alterations of cardiac microenvironment of diabetic mice.

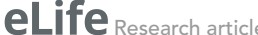

**Figure 2.** Comparison analysis of the communications between cardiac cells during high-fat diet (HFD)/streptozotocin (STZ)-induced diabetes. (**A**) Heatmap showing the enriched Gene Ontology (GO) terms associated with extracellular matrix remodeling and myocardial fibrosis in major cardiac cell populations in the diabetic group. (**B**) The relative expression of selected essential growth factors in the major cardiac cell types. (**C**) Heatmap showings the number of ligand–receptor pairs between cardiac cell populations in healthy mice. (**D**) Heatmap shows the number of ligand–receptor pairs between cardiac cell populations in the HFD/STZ-induced diabetic mice. (**E**) Bar plot shows total number of connections made by each cell type without (gray bars) and with HFD/STZ treatment (red bars). (**F**) Bar plot illustrates the number of upregulated receptors and ligands for each population of cardiac cells. (**G**) Bar plot showing the number of downregulated receptors and ligands in each cardiac cell population. DB, diabetes. The details of unique differentially expressed genes (uni-DEGs) in the cardiac populations are listed in **Supplementary file 4**. The details of significantly differentially expressed genes in the specific cell populations relative to others in mouse heart are listed in **Supplementary file 5**. Details of cell type-specific receptors in the cardiac populations and cell type-specific ligands in cardiac populations are listed in **Supplementary file 6** and **Supplementary file 7**, respectively. The details of relative expression of a selection of essential growth factors across major cardiac cell types are listed in **Supplementary file 8**. The details of the number of ligand–receptor pairs between cardiac cell populations in healthy mice or diabetic mice are listed in **Supplementary file 9** and **Supplementary file 10**, respectively. The details of significantly differentially expressed ligands and receptors in each cell population are listed in **Supplementary file 11** and **Supplementary file 12**, respectively.

The online version of this article includes the following figure supplement(s) for figure 2:

**Figure supplement 1.** The expression of receptors and ligands across major cardiac cell types.

## Identification of key ligand–receptor pairs associated with diabetic myocardial fibrosis in fibroblasts

Cardiac fibroblasts are the primary drivers of myocardial fibrosis (*Travers et al., 2016*; *Frangogiannis, 2021*). Given the result that cardiac fibroblasts cause the greatest increase in the number of connections in response to HFD/STZ-induced diabetes, perturbations of intercellular communications between cardiac fibroblasts and cardiac populations may be key drivers of diabetic myocardial fibrosis. To investigate the key receptor–ligand interactions in diabetic myocardial fibrosis, highly expressed receptors in cardiac fibroblasts were screened (*Figure 3—figure supplement 1A*, FDR ≤0.01, log2FC ≥1). Then, we merged the upregulated uni-DEGs in the fibroblasts and the highly expressed fibroblast receptors, whose cognate ligands were upregulated in at least one cardiac cell type during diabetic progression (*Figure 3A*). A protein–protein interaction (PPI) network was constructed using the new fibroblast-specific gene set (*Figure 3B*). *Egfr* and *Pdgfra* were among the top hub genes based on the node degree that were specifically high-expressed receptors in the cardiac fibroblasts (*Figure 3—figure supplement 1B, C*). To clarify their role in fibrotic progression, we screened for the cognate ligands of *Egfr* and *Pdgfra* in each cardiac cell population (*Figure 3—figure supplement 2A, B*, *Supplementary file 13*, and *Supplementary file 14*). Both *Pdgfb* and *Pdgfd* transcripts were upregulated in endothelial cells, while *Pdgfc* levels were significantly elevated in the macrophages of diabetic mice (*Figure 3C, D* and two-sided Wilcoxon rank-sum test, FDR ≤0.05, log2FC ≥0.36). The protein levels of Pdgfb, Pdgfd, and Pdgfc showed similar changes in the corresponding cardiac cells (*Figure 3—figure supplement 1D–F*, $n = 6$ mice per group). In addition, the *Egfr* ligand *Efemp1* was upregulated in the epicardial cells (*Figure 3E* and two-sided Wilcoxon rank-sum test, FDR ≤0.05, log2FC ≥0.36). These results suggest that the interactions between cardiac fibroblasts with endothelial cells, macrophages and epicardial cells through Pdgf(s)–Pdgfra and Efemp1–Egfr may contribute to myocardial fibrosis in the diabetic mouse heart.

Pdgfra exerts its tyrosine kinase activity by binding to its cognate ligands. Immunostaining of cardiac tissue revealed significantly higher protein levels of Pdgfb, Pdgfc, and Pdgfd in the Pdgfra-positive cells of the diabetic group (*Figure 3—figure supplement 3A–C*, $n = 6$ mice per group). To examine the functional role of Pdgfra in diabetic myocardial fibrosis, we treated the HFD/STZ-induced diabetic mice with the Pdgfra inhibitor imatinib mesylate (Ima). The results showed that HFD/STZ treatment significantly increased cardiac p-Pdgfra protein levels and decreased that of p-Pdgfra in the HFD/STZ + Ima group compared with the HFD/STZ group (*Figure 3F*, $n = 6$ mice per group). In addition, the myocardium of the HFD/STZ-treated mice expressed higher levels of *Col1a1* and *Col3a1* compared with the control group (*Figure 3G, H*, $n = 6$ mice per group, mean ± standard error of mean [SEM], **$p < 0.01$, ***$p < 0.001$), which coincided with increased collagen deposition in the extracellular matrix. However, Ima treatment attenuated the increase in HFD/STZ-induced collagen synthesis and deposition (*Figure 3I*, $n = 6$ mice per group). Next, we performed WGA staining to assess myocardial hypertrophy in mice. The myocardial hypertrophy of HFD/STZ diabetic mice was improved significantly after Ima treatment (*Figure 3—figure supplement 4*, $n = 6$ mice per group). Finally, we evaluated the improvement of heart function in diabetic mice after Ima treatment using echocardiography (*Figure 3—figure supplement 5*, $n = 15$ mice per group, mean ± standard deviation; *$p < 0.01$; **$p < 0.01$; ***$p < 0.001$). As shown in the *Figure 3—figure supplement 5*, Ima treatment significantly improved the LV function of the diabetic mice as evaluated using IVSs, EF, FS, LV mass, and the E/A ratio. Taken together, our results show that Pdgf(s)–Pdgfra interactions contribute to diabetic myocardial fibrosis.

## Identification of myocardial fibrosis-related cardiac fibroblast subpopulation

Cell subpopulations in the tissues perform non-overlapping functions and play different biological roles (*Croft et al., 2019*). Cell types can be defined by the unbiased clustering of single cells based on global transcriptome patterns (*Rozenblatt-Rosen et al., 2017*; *McLellan et al., 2020*). To observe the heterogeneity of fibroblasts in the heart, we examined 6416 fibroblasts from the diabetes and control mice. Unsupervised Seurat-based clustering of the 6416 fibroblasts revealed 10 distinct subpopulations (*Figure 4A*, *Supplementary file 15*, and $n = 3428$ fibroblasts from healthy control and $n = 2988$ fibroblasts from 6 diabetic mice). Next, hierarchical clustering with multiscale bootstrap resampling was used to analyze the heterogeneity of these cardiac fibroblast subpopulations. The analysis



**Figure 3.** Identification of key ligand–receptor pairs associated with diabetic myocardial fibrosis in the fibroblasts. (**A**) Heatmap showing the pairs of highly expressed fibroblast receptors and upregulated ligands in each cell type in the diabetic hearts. (**B**) PPI network showing the interactions of the upregulated genes in the fibroblasts. The circle size represents the protein node degree in the network. Volcano plots of the DEGs in the heart tissues of high-fat diet (HFD)/streptozotocin (STZ)-treated and control mice. *Pdgfa*, *Pdgfb*, *Pdgfc*, and *Pdgfd* expression in the endothelial cells (**C**) and macrophages (**D**) are highlighted. (**E**) Volcano plots of the DEGs in the hearts of the HFD/STZ-treated and control mice. *Efemp1* expression in the epicardial cells is highlighted. (**F**) Representative immunofluorescence images of p-Pdgfra in the heart tissues of the HFD/STZ-treated mice with or without Ima treatment (*n* = 6 mice per group), scale bar = 40 μm. Bar plots showing mRNA expression of *Col1a1* (**G**) and *Col3a1* (**H**) in the heart tissues of the HFD/STZ-treated mice with or without Ima treatment (*n* = 6 mice per group; mean ± SEM; **p < 0.01; ***p < 0.001). (**I**) Representative images of Masson dye-stained heart sections from the groups indicated showing the extent of collagen deposition (*n* = 6 mice per group), scale bar = 20 μm. Ima, imatinib mesylate; SEM, standard error of mean. The details of the cognate ligands of *Egfr* and *Pdgfra* are listed in **Supplementary file 13** and **Supplementary file 14**, respectively.

*Figure 3 continued on next page*

*Figure 3 continued*

The online version of this article includes the following source data and figure supplement(s) for figure 3:

**Source data 1.** Source data for CT values of *Col*1a1 used for *Figure 3G*.

**Source data 2.** Source data for CT values of *Col*3a1 used for *Figure 3H*.

**Figure supplement 1.** Identification of key ligand–receptor pairs associated with diabetic myocardial fibrosis in the fibroblasts.

**Figure supplement 2.** The cognate ligands of *Egfr* in each cardiac cell population.

**Figure supplement 3.** Immunofluorescence results of Pdgfb, Pdgfc, and Pdgfd in the Pdgfra⁺ cells.

**Figure supplement 4.** Assessment of myocardial hypertrophy using by WGA staining.

**Figure supplement 5.** Echocardiographic evaluation of mice heart function.

**Figure supplement 5—source data 1.** Source data files are provided to support Table 1 in *Figure 3—figure supplement 5*.

revealed that fibroblast subpopulation 3 formed a distinct cluster from other fibroblast subpopulations, based on its expression pattern (*Figure 4B*).

We further investigated the contribution of all fibroblast populations to myocardial fibrosis. The top 5 ranking markers in the heart showed distinct signatures for each subpopulation of fibroblasts using heatmap analysis (*Figure 4C*, *Supplementary file 16*, FDR ≤0.05, log2FC ≥0.36). It is worthy to note that the top enriched genes in subpopulation 3 (*Nppa* and *Clu*) and subpopulation 5 (*Postn* and *Cilp*) are well-established biomarkers of pro-fibrotic function. The gene set variation analysis (GSVA) of each fibroblast subpopulation suggested a diversification of function between the subpopulations, and fibroblast 3 and 5 populations were strongly involved in extracellular matrix remodeling and collagen synthesis (*Figure 4D*, FDR ≤0.05). These results indicate that fibroblast 3 and 5 subpopulations are myocardial fibrosis-related cardiac fibroblast subpopulations.

The most significantly enriched gene in subpopulation 3 was *Hrc*, which is crucial for proper cardiac function through the regulation of Ca²⁺-uptake, storage, and release. The most significantly enriched gene in fibroblast subpopulation 5 was *Postn*, which is consistent with the fibroblast subset identified in an animal model of angiotensin-induced myocardial hypertrophy (*McLellan et al., 2020*). The *Hrc*ʰⁱ and *Postn*ʰⁱ fibroblast populations were also detected in mouse heart by immunostaining for Hrc and Postn, respectively (*Figure 4E, F*, *n* = 6 mice per group).

## Transcription factor network analysis

To investigate the underlying molecular mechanisms that drive the phenotypic differentiation of the fibroblast subpopulations, we used single-cell regulatory network inference and clustering. The results revealed the upregulation of different transcription factor networks in each subpopulation. For instance, *Thra* and *Creb5* regulons were upregulated in fibroblast subpopulation 0 and 2, respectively, whereas the *Nfe2l1* network was enriched in subpopulation 3, while subpopulation 4 showed the increased activation of the *Foxp2* network (*Figure 5A*, *Supplementary file 17*). Regulons driven by *Tcf4* transcription factors were enriched in subpopulations 1 and 9, while *Mef2a* was enriched in subpopulations 7 and 8. Consistent with the role of *Hmgb1* in cardiac fibrosis (*Wu et al., 2018*), a *Hmgb1*-based network was upregulated in the *Hrc*ʰⁱ and *Postn*ʰⁱ fibroblast populations (*Figure 5A*). Heatmap analysis further confirmed the upregulation of transcription factors (*Figure 5B*, *Supplementary file 18*).

## Identification of intercellular communication drivers of myocardial fibrosis in the *Hrc*ʰⁱ fibroblasts

To identify the cellular drivers of myocardial fibrosis during diabetes, we performed a cluster analysis of the DEGs between control and diabetic mice heart. HFD/STZ treatment induced transcriptional changes in all cardiac fibroblast subpopulations (*Figure 6—figure supplement 1*, *Supplementary files 19*, and two-sided Wilcoxon rank-sum test, FDR ≤0.05, log2FC ≥0.36), and the upregulated genes in the *Hrc*ʰⁱ fibroblasts were enriched in GO terms such as collagen fiber reorganization and extracellular matrix binding (*Figure 6A* and two-sided Wilcoxon rank-sum test, FDR ≤0.05). Furthermore, the top 20 enriched pathways in the *Hrc*ʰⁱ fibroblasts of the diabetic group were associated with extracellular matrix organization, myocardial fibrosis, and fibroblast activation (*Figure 6B* and

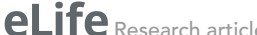

**Figure 4.** Analysis of the heterogeneity of the fibroblast subpopulations. (**A**) t-SNE plot of the 10 cardiac fibroblast subpopulations in the high-fat diet (HFD)/streptozotocin (STZ)-treated and control mice (n = 3428 fibroblasts from healthy control and n = 2988 fibroblasts from 6 diabetic mice). (**B**) Correlation heatmap of the gene-expression signatures of each fibroblast subpopulation. Color differences indicate subpopulations that were resolved through multiscale bootstrapping. (**C**) Heatmap showing the top 5 marker genes in each fibroblast subpopulation. The red color indicates high expression, while green color indicates low expression. (**D**) Heatmap showing the enriched Gene Ontology (GO) terms associated with cardiac fibrosis in each fibroblast population. (**E, F**) Representative immunofluorescence images of Hrc (**E**) and Postn (**F**) in mouse heart (n = 6 mice per group), scale bar = 100 μm. The details of the 10 transcriptionally distinct fibroblast subpopulations are listed in **Supplementary file 15**. The details of distinct signatures of each fibroblast subpopulation in the heart are listed in **Supplementary file 16**.

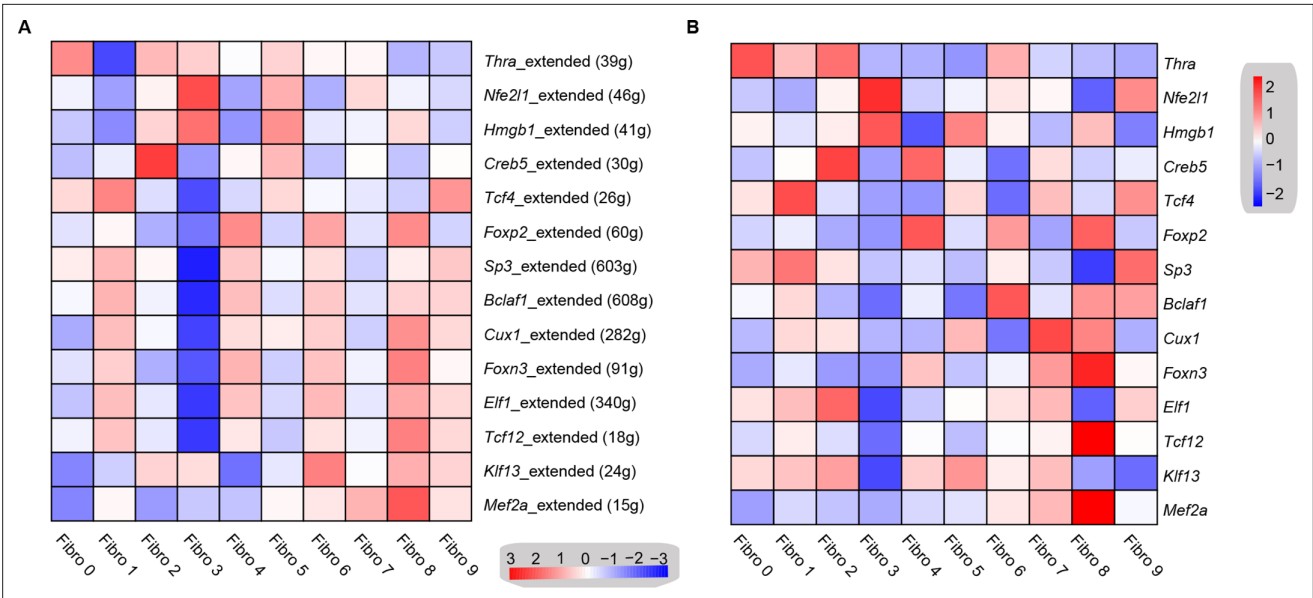

**Figure 5.** Transcription factor network analysis of fibroblast subpopulations. (**A**) Heatmap shows the inferred transcription factor gene-regulatory networks. Numbers between brackets indicate the (extended) regulons for respective transcription factors. (**B**) Heatmap shows the expression level of transcription factors in (**A**). The details of transcription factor gene-regulatory networks in the distinct subpopulations are listed in **Supplementary file 17**. The details of the transcription factors expression are listed in **Supplementary file 18**.

two-sided Wilcoxon rank-sum test, FDR ≤0.05). These results suggest that the $Hrc^{hi}$ fibroblasts are key cellular drivers of myocardial fibrosis in response to HFD/STZ-induced diabetes.

To identify the critical signaling molecules in the $Hrc^{hi}$ fibroblasts that mediate myocardial fibrosis during diabetics, we identified the uni-DEGs in each fibroblast subpopulation (**Supplementary file 20**) and constructed a PPI network using the upregulated genes (**Figure 6C**). The top 15 hub genes included *Itgb1*, *Col6a1*, *Col1a2*, *Dcn*, *Rpl6*, *Rps20*, *Serpinh1*, *Bgn*, *Hsp90aa1*, and *Col6a2*, of which *Col6a1*, *Col1a2*, *Col6a2*, *Dcn*, and *Bgn* encode for ECM proteins (**Schipke et al., 2017**). In addition, both *Serpinh1* and *Hsp90aa1* have been reported to participate in collagen synthesis (**Christiansen et al., 2010**; **García et al., 2016**). Although the role of these candidate hub genes has been well established in myocardial fibrosis, the function of *Itgb1* is currently unknown.

The PPI network of the upregulated uni-DEGs and immunostaining results indicate the key role of *Itgb1* of $Hrc^{hi}$ fibroblasts in diabetic myocardial fibrosis (**Figure 6D**, n = 6 mice per group; **Figure 6—figure supplement 2**, two-sided Wilcoxon rank-sum test, FDR ≤0.05, log2FC ≥0.36). *Itgb1* deficiency increases the risk of ventricular arrhythmias in patients with arrhythmogenic right ventricular cardiomyopathy (**Wang et al., 2020**). It is therefore reasonable to surmise that the interaction between *Itgb1* and its cognate ligand is involved in diabetes-related myocardial fibrosis. To confirm this hypothesis, we screened all potential ligands of *Itgb1* (**Supplementary file 21**), and found that Lgals3bp and Fn1 were upregulated in the heart tissues of diabetic mice (**Figure 6E - G**, n = 6 mice per group).

To validate the role of *Itgb1* in the heart in myocardial fibrosis during diabetes, we used the adeno-associated virus 9 (AAV9) to deliver *Itgb1* siRNA, which preferentially targets the heart. The level of *Itgb1* mRNA decreased by >80% in response to a single injection of *Itgb1* siRNA, compared with the negative control (**Figure 6H**, n = 6 mice per group, mean ± SEM, ****p < 0.0001). Moreover, the knockdown of *Itgb1* lasted for more than 5 months after the injection of the siRNA. Next, we tested collagen synthesis and deposition in the diabetic mouse heart with *Itgb1* knockdown using Masson dye staining. The results showed that levels of collagen synthesis and deposition were indeed downregulated in the *Itgb1* knockout mice (**Figure 6I**, n = 6 mice per group). Taken together, our results show that *Itgb1* in $Hrc^{hi}$ fibroblasts contributes to HFD/STZ-induced myocardial fibrosis. Further studies are warranted to establish the exact function of these ligand–receptor interactions in diabetic myocardial fibrosis.



**Figure 6.** Identification of intercellular communication drivers of myocardial fibrosis in the *Hrc*<sup>hi</sup> fibroblasts. (**A**) Heatmap showing high-fat diet (HFD)/streptozotocin (STZ)-induced enrichment of Gene Ontology (GO) terms associated with extracellular matrix remodeling and myocardial fibrosis in each subpopulation of cardiac fibroblasts. (**B**) Dot plot of the GO analysis showing the top 20 terms with the highest enrichment in the *Hrc*<sup>hi</sup>

*Figure 6 continued on next page*

*Figure 6 continued*

fibroblasts of the HFD/STZ-treated mice relative to controls. (**C**) PPI network showing the interactions of the upregulated genes in the *Hrc^hi* fibroblasts of diabetic mice relative to controls. The circle size represents the protein node degree in the network. (**D**) Representative immunofluorescence images of Itgb1 in the heart of SHH-fed or control mice (*n* = 6 mice per group). scale bar = 40 µm. (**E**) Volcano plots showing the DEGs in the heart tissues of the HFD/STZ-treated or control mice with *Fn1* and *Lgals3bp* highlighted. Representative immunofluorescence images for Fn1 (**F**) and Lgals3bp (**G**) in the mouse heart (*n* = 6 mice per group), scale bar = 100 µm. (**H**) The efficiency of siRNA-mediated *Itgb1* mRNA knockdown was confirmed by qRT–PCR (*n* = 6 mice per group, mean ± standard error of mean [SEM], ****p < 0.0001). (**I**) Representative images of Masson dye-stained heart sections from the indicated groups showing extent of collagen deposition (*n* = 6 mice per group), scale bar = 20 µm. Detailed genes of significant transcriptomic changes in each fibroblast subpopulation are listed in *Supplementary file 19*. The details of unique differentially expressed genes (uni-DEGs) in each fibroblast subpopulation are listed in *Supplementary file 20*. The details of the cognate ligands of *Itgb1* are listed in *Supplementary file 21*.

The online version of this article includes the following source data and figure supplement(s) for figure 6:

**Source data 1.** Source data for CT values of *Itgb1* used for *Figure 6H*.

**Source data 2.** Source data for CT values of *Itgb1* used for *Figure 6H*.

**Figure supplement 1.** Up- and downregulated genes in each fibroblast subpopulation of diabetic mice, compared with the control mice.

**Figure supplement 2.** DEGs in each cardiac fibroblast subpopulation from high-fat diet (HFD)/streptozotocin (STZ)-treated or control mice with *Itgb1* highlighted.

## Validating the results of scRNA-seq data in the DB/DB diabetic mice model

To confirm if the scRNA-seq data is consistent with that of other mouse models of type 2 diabetes, we performed a data validation in the DB/DB mice. DB/DB is a spontaneous transgenic mouse model of type 2 diabetes with leptin receptor point mutation, which develops symptoms of obesity, insulin resistance, and hyperglycemia. We found that Pdgfra phosphorylation was significantly promoted in the DB/DB mice, and the phosphorylation level of Pdgfra was significantly inhibited after Ima treatment (*Figure 7A*, *n* = 6 mice per group). Subsequently, we examined whether Hrc-activated fibroblasts could be detected in the DB/DB mice, and we performed Hrc and Desmin co-staining experiment. Hrc was highly expressed in DB/DB diabetic mice and overlaps with the fibroblast marker Desmin, suggesting that Hrc is significantly activated in the fibroblasts of the diabetic heart. After treatment with Ima, the expression of Hrc and Desmin was significantly inhibited (*Figure 7B*, *n* = 6 mice per group). Finally, the qRT-PCR results of *Acta2* and *Col3a1* further confirmed that Ima treatment could significantly inhibit the accumulation of collagen fibers in the heart of the diabetic mice (*Figure 7C, D*, *n* = 4 mice for *Acta2* and *n* = 6 mice for *Col3a1*).

Next, we also examined the role of the Pdgf(s)–Pdgfra axis in the fibrogenesis of diabetic myocardium, and confirmed that Pdgf(s)–Pdgfra axis-mediated ligand–receptor interaction play a similar important role in the fibrogenic progression of the myocardium in DB/DB diabetic mice through the double immunofluorescence staining of Pdgfra with Pdgfc and Pdgfd (*Figure 7—figure supplement 1*, *n* = 6 mice per group).

## Identification of the results of scRNA-seq data in female C57 diabetic mice

To further verify the results of scRNA-seq in diabetic cardiomyopathy, we performed validation studies on female C57 mice. First, we examined whether Pdgfra is activated in the heart of diabetic mice by performing p-Pdgfra and desmin double staining, and the results showed that Pdgfra phosphorylation was significantly activated in the diabetic mice, and that phosphorylation was inhibited after Ima treatment (*Figure 8A*, *n* = 6 mice per group). Subsequently, we examined whether there was a subpopulation of *Hrc^+* fibroblasts in the hearts of the female diabetic mice. Hrc is highly expressed in the heart of female mice and overlaps with that of the cardiac fibroblast marker, desmin. However, Ima treatment significantly inhibited the activation of Hrc fibroblasts (*Figure 8B*, *n* = 6 mice per group). These results suggest that the appearance of *Hrc^+* fibroblast subsets is a common phenomenon in

**Figure 7.** Identification of Hrc-activated fibroblasts and Pdgfra phosphorylation in DB/DB diabetic mice. (**A**) p-Pdgfra (green) and Desmin (red) labeled fibroblasts in the mice heart. (**B**) Hrc (green) and Desmin (red) labeled activated fibroblasts in the mice heart (n = 6 mice per group for A and B; scale bar = 50 μm). (**C**) *Acta2* mRNA expression level in the mice heart (n = 4 mice per group; mean ± standard error of mean [SEM]; *p < 0.05; ***p < 0.001). (**D**) *Col3a1* mRNA expression level in mice heart (n = 6 mice per group; mean ± SEM; **p < 0.01; ***p < 0.001).

*Figure 7 continued on next page*

*Figure 7 continued*

The online version of this article includes the following figure supplement(s) for figure 7:

**Figure supplement 1.** Immunostaining results of Pdgfd and Pdgfc in the Pdgfra⁺ fibroblasts.

diabetic cardiomyopathy. Further examination of *Acta2* (*Figure 8C*, *n* = 4 mice per group) and *Col3a1* (*Figure 8D*, *n* = 4 mice per group) mRNA levels revealed that Ima treatment significantly inhibited collagen accumulation in the heart of diabetic mice.

Finally, we further examined whether the Pdgf(s)–Pdgfra interactions are involved in the pathogenesis of diabetic myocardial fibrosis. The results suggest that ligand–receptor interactions between macrophages, endothelial cells, and fibroblasts play an important role in promoting myocardial fibrosis (*Figure 8—figure supplement 1*, *n* = 6 mice per group). This theory has been validated in both male C57 mice and DB/DB diabetes model and appears to be a common mechanism that promotes cardiac fibrosis in diabetes.

## Discussion

scRNA-seq allows for the in-depth analysis of individual cells in heterogenous populations (*Butler et al., 2018*) in healthy and diseased tissues (*Ackers-Johnson et al., 2018*; *Mathys et al., 2019*; *Peng et al., 2019*; *Kalucka et al., 2020*; *Litviňuková et al., 2020*; *Li et al., 2021*). However, the effect of diabetes on cardiac cell function and cardiac cell heterogeneity at single-cell level has not been previously reported. In this study, we mapped out the transcriptional alterations associated with HFD/STZ-induced diabetes in different cardiac cell populations, and identified the key ligand–receptor pair drivers of myocardial fibrosis in the fibroblasts of the diabetic heart. Specifically, the emergence of *Hrc^hi* fibroblast subpopulations in response to diabetic progression, presumably to remodel the extracellular environment through multiple ligand–receptor interactions. In addition, the appearance of the *Postn*⁺ fibroblast subpopulation was also identified in this study. Interestingly, a subset of *Postn*⁺ fibroblasts was found in both human dilated cardiomyopathy and mouse cardiac hypertrophy models (*Koenig et al., 2022*; *McLellan et al., 2020*), and our scRNA-seq results also identified *Postn*⁺ fibroblasts. These results suggest that *Postn*⁺ fibroblasts may play a similar important role in the various pathology of cardiac fibrosis.

The heart of a mammal is a complex organ composed of a variety of cell types (*Banerjee et al., 2007*; *Litviňuková et al., 2020*). Cardiac fibroblasts synthesize extracellular matrix proteins and their excessive activation in response to stress induced cardiac fibrosis (*Travers et al., 2016*). GO enrichment analysis of all upregulated genes in the cardiac cell populations from the diabetic mice confirmed the association of fibroblasts and extracellular matrix remodeling with myocardial fibrosis, which is consistent with the results of previous studies (*Jia et al., 2018*; *Ivey et al., 2019*; *McLellan et al., 2020*; *Frangogiannis, 2021*; *Koenig et al., 2022*). The survival and proper functioning of metazoans depends on communication between multiple cell populations and tissues via secretary ligands and membrane receptors (*Ramilowski et al., 2015*). Therefore, we screened for receptor genes that were highly expressed in the cardiac fibroblasts of diabetic mice and their cognate ligands that were upregulated in other cardiac cell populations to identify the dysregulated ligand–fibroblast receptor interactions. Protein–protein interaction network analysis indicated that the receptors, *Pdgfra* and *Egfr*, which are highly expressed in the fibroblasts, play a central role in myocardial fibrosis in diabetes. Pdgfra is a surface receptor tyrosine kinase that is activated upon binding to its corresponding ligand Pdgf(s), and regulates cell division and proliferation (*Rudat et al., 2013*; *Gouveia et al., 2018*; *Soliman et al., 2020*). Egfr on the other hand is a member of the ErbB family of receptor tyrosine kinases and plays an important role in wound healing and cardiac hypertrophy (*Peng et al., 2016*). The upregulation of the Pdgfra ligands, Pdgfb and Pdgfd in endothelial cells and Pdgfc in macrophages, and that of the Egfr ligand, Efemp1 in epicardial cells of the HFD/STZ-treated mice indicate that the cardiac microenvironment had changed, resulting in extracellular matrix remodeling and cardiac fibrosis. This is of particular interest given the pathological role of these cell populations and ligand–receptor pairs in cardiovascular diseases (*Rottlaender et al., 2011*; *Shinagawa and Frantz, 2015*; *Farbehi et al., 2019*; *Marín-Juez et al., 2019*; *Peet et al., 2020*; *Baguma-Nibasheka et al., 2021*). Ligand–receptor pair analysis also revealed the synergistic role of endothelial cells, macrophages, and epicardial cells

**Figure 8.** Validation of Hrc-activated fibroblasts and Pdgfra phosphorylation in female C57 mice. (**A**) p-Pdgfra (green) and Desmin (red) labeled fibroblasts in the mice heart. (**B**) Hrc (green) and Desmin (red) labeled activated fibroblasts in the mice heart. *n* = 6 mice per group for A and B; scale bar = 50 µm. (**C**) *Acta2* mRNA expression level in the mice heart (*n* = 4 mice per group; mean ± standard error of mean [SEM], *p < 0.05; ***p < 0.001). (**D**) *Col3a1* mRNA expression level in mice heart (*n* = 4 mice per group; mean ± SEM; **p < 0.01; ***p < 0.001).

*Figure 8 continued on next page*

*Figure 8 continued*

The online version of this article includes the following figure supplement(s) for figure 8:

**Figure supplement 1.** Immunofluorescence results of Pdgfd and Pdgfc in the Pdgfra⁺ fibroblasts.

with fibroblasts in diabetic myocardial fibrosis. Further analysis of the interactions between these cell populations will help us understand the pathogenesis of diabetes-induced fibrosis.

Terminally differentiated cells are generally considered to have limited plasticity. Cellular plasticity in adults is mostly reported during the terminal differentiation stage of many progenitor cells (*Chang-Panesso and Humphreys, 2017*). However, these cellular transitions may also be present in cardiac fibroblasts. Unbiased single-cell clustering can redefine cell type based on the global transcriptome patterns (*Rozenblatt-Rosen et al., 2017*; *Ackers-Johnson et al., 2018*; *McLellan et al., 2020*; *Koenig et al., 2022*). Such analyses have already been applied to other organs (*Macosko et al., 2015*; *Chen et al., 2017*; *Stubbington et al., 2017*) and even to whole multicellular organisms (*Cao et al., 2017*; *Karaiskos et al., 2017*). These experiments have identified new cells as well as previously defined cells with catalogued marker genes, demonstrating that this approach can be used to redefine cardiac cell types. One of the most important results of the mouse cardiac fibroblasts analysis was the identification of two different phenotypic $Hrc^{hi}$ and $Postn^{hi}$ fibroblast subpopulations associated with extracellular matrix remodeling. The Postn^hi fibroblasts participate in fibrogenic progression, which is consistent with another subpopulation of fibroblasts identified in an animal model of angiotensin-induced myocardial hypertrophy (*McLellan et al., 2020*; *Koenig et al., 2022*), suggesting that these fibroblasts may contribute to both myocardial hypertrophy and cardiac fibrosis. $Hrc^{hi}$ fibroblasts express fibrogenic marker genes, such as *Nppa*, *Ttn*, and *Clu*, which indicates a key pro-fibrotic function. *Hrc* knockout or AAV-mediated knockdown results in pulmonary edema, severe cardiac hypertrophy, fibrosis, heart failure, and decreased survival after transverse aortic constriction in mice (*Park et al., 2012*; *Park et al., 2013*). Combined with our single-cell sequencing results, we can surmise that *Hrc* is a potential target for the inhibition of myocardial fibrosis during diabetes.

Strikingly, GSVA and GO analyses of each fibroblast subpopulation indicated that the $Hrc^{hi}$ fibroblasts were the most profibrogenic under diabetic conditions. This finding suggests that the $Hrc^{hi}$ fibroblasts may be key cellular drivers of myocardial fibrosis in diabetes. We speculated that intercellular communication between $Hrc^{hi}$ fibroblasts and other cardiac cells is a constitutive process activated in diabetes. Intercellular and protein-protein interaction networks within the $Hrc^{hi}$ fibroblasts reveal the key role of the receptor, *Itgb1*, in diabetic myocardial fibrosis. The potential ligands of Itgb1, Lgals3bp, and Fn1, are upregulated in the heart tissues of diabetic mice. Perhaps, the gain-of-function of the Lgals3bp–Itgb1 and Fn1–Itgb1 pairs may explain the role of $Hrc^{hi}$ fibroblasts in diabetic myocardial fibrosis.

In summary, we mapped the transcriptional alterations associated with HFD/STZ-induced diabetes in different cardiac subpopulations, and identified key ligand–receptor pair drivers of myocardial fibrosis in the diabetic heart, specifically the Pdgf(s)–Pdgfra and Efemp1–Egfr interactions, which are mediated by fibroblasts with macrophages, endothelial cells, and epicardial cells. Crucially, $Hrc^{hi}$ fibroblasts were identified as the key profibrogenic subpopulation that may contribute to cardiac fibrosis by remodeling the extracellular environment through drivers of intercellular communication mediated by Itgb1. Therefore, we speculate that the specific targeting of $Hrc^{hi}$ fibroblasts will be a promising target for the treatment of myocardial fibrosis. Our future research direction will involve the combination of fibroblast-specific *Hrc* knockout mice and the analysis of cardiac cellular networks to confirm the role of *Hrc* in regulating diabetic myocardial fibrosis.

## Materials and methods
### Animals and treatments

Male and female C57BL/6J mice weighing 18–22 g that were 6 weeks old were purchased from the Center for Laboratory Animals, Soochow University. After 1 week of acclimatization, a diabetic mouse model was constructed as previously described with some modifications (*Lu et al., 2011*; *Li et al., 2017*). The mice in the normal group were fed on a normal diet, while all mice in the other groups were fed with HFD (60% fat, 20% protein, and 20% carbohydrate) during the animal experiments.

After 6 weeks of HFD feeding, the mice in the diabetic control model and imatinib group were fasted for 12 hr every night and injected with STZ (35 mg/kg, dissolved in 0.1 mM cold citrate buffer, pH 4.4) for 3 days to induce diabetes. Meanwhile, the control group was injected with a citrate buffer. After a week of STZ injection, the 12 hr fasting glucose level of all mice was measured. Mice with fasting blood glucose levels of ≥11.1 mmol/l (*Yu et al., 2014*) were considered to be type 2 diabetes mice. Then, imatinib (40 mg/kg) was administered daily through intraperitoneal injection during the procedure. After 21 weeks of the injection of STZ, the mice were anesthetized using an intraperitoneal injection of pentobarbital sodium. The hearts were dissected and stored at −80°C for further analysis. During the experiment, the mice were kept on their respective diets and their body weight was measured weekly. All procedures were performed with minimal damage to the mice. DB/DB and DB/M transgenic mice that were 8 weeks old were purchased from the Center for Laboratory Animals, Soochow University. For the DB/M normal control group, the mice were fed on a normal diet, while the diabetic DB/DB mice were divided into two groups, one was diabetic control group and the other was Ima treatment group. After 8 weeks of injection of Ima, all mice were euthanized and the hearts were dissected for further analysis.

## Echocardiographic evaluation

We performed echocardiographic measurements before and after the experimental intervention. A Vevo 2100TM High Resolution Imaging System (Visual Sonics) was used for M-mode echocardiography. According to a previous study, an evaluation of the following structural variables was conducted: left ventricular internal dimension in diastole (LVIDD), left ventricular internal dimension in systole (LVIDS), interventricular septal thickness in diastole (IVSd) and in systole (IVSs), and LV posterior wall thickness in diastole (LVPWd) and systole (LVPWs). Ejection fraction (EF), fractional shortening (FS), and E/A ratio were used to evaluate left ventricular (LV) function. We calculated the LV mass using the formula [(LVIDd + LVPWd + IVSd)3 − (LVIDd)3 × 1.04 × 0.8 + 0.6].

## Immunofluorescence

After fixing in 4% paraformaldehyde(PFA) and dehydrating with 20% sucrose, the hearts were embedded in the optimal cutting temperature compound and stored at −80°C. They were then sectioned using a Leica CM1950 system into 10-μm-thick horizontal slices. The sections were incubated with the primary antibody (anti-CD68 (Abcam, ab955), anti-CD31 (Abcam, ab28364), anti-CD31 (BD, 553700), anti-Pdgfb (CST, 3169T), anti-Pdgfc (Abcam, ab200401), anti-Pdgfd (Abcam, ab234666), anti-Pdgfra (R&D, AF1062-SP), anti-phospho-Pdgfra (Tyr754) (Thermo Fisher, 441008G), anti-Vim (R&D, BAM2105), anti-Hrc (Proteintech, 18142-1-AP), anti-Postn (R&D, AF2955-SP), anti-Itgb1 (Invitrogen, 14-0299-82), anti-FN1 (Abcam, ab2413), and anti-Lgals3bp (Abcam, ab236509)) or an IgG control for the immunofluorescence staining. The fluorescent secondary antibodies (goat anti mouse IgM Alexa Fluor 647, abcam, ab150123, or donkey anti rabbit IgG Alexa Fluor 568, abcam, ab175470) and DAPI (SouthernBiotech, 0100-01) were used to visualize the specific proteins. Wheat Germ Agglutinin (WGA) kit was obtained from (Thermo Fisher, W11261).

## Real-time quantitative polymerase chain reaction

Total RNA from the mouse hearts was extracted using QIAGEN's miRNeasy Mini kit (217004; Qiagen, Germany). The reverse transcription step was performed using Takara's PrimeScript RT Master Mix (RR036A; Takara, Japan). A brilliant SYBR green PCR master mix (4913914, Roche, Switzerland) was used to perform qPCR on the cDNA templates in a LightCycler 480 system (Roche, Switzerland). The target mRNA expression levels were normalized to that of *Gapdh* and the relative fold change was calculated using the $2^{-\Delta\Delta CT}$ method. The qPCR primers for *Col1a1* (forward 5'- AACTCCCTCCAC CCCAATCT, reverse 5'-CCATGGAGATGCCAGATGGTT), *Col3a1* (forward 5'-ACGTAAGCACTGGTGG ACAG, reverse 5'-GGAGGGCCATAGCTGAACTG), *Itgb1* (forward 5'-ATGCCAAATCTTGCGGAG AAT, reverse 5'-TTTGCTGCGATTGGTGACATT), and *Gapdh* (forward 5'-GGTCATCCATGACAACTT, reverse 5'-GGGGCCATCCACAGTCTT) were designed and synthesized by Invitrogen (Shanghai, China).

## siRNA-mediated knockdown of *Itgb1* in the mice heart

The siRNAs targeting *Itgb1* were used, as previously reported (*Speicher et al., 2014*). Sense 5'-AGAuGAGGuucAAuuuGAAdTsdT, antisense 5'-UUcAAAUUGAACCUcAUCUdTsdT. Negative

control: sense 5′-cuuAcGcuGAGuAcuucGAdTsdT, antisense 5′-UCGAAGuACUcAGCGuAAGdTsdT. These targeted siRNA sequences were subcloned into an AAV9 plasmid vector and packaged into a AAV virus particle in vitro, and the titers of the AAV viruses were ensured to exceed $1 \times 10^{12}$ vg/ml. Diabetic C57BL/6 male mice received the negative control or *Itgb1* siRNA mediated by AAV (0.5 mg/ kg) via tail vein injection at a volume of 5 ml/kg body weight on days 1 and 5 after the first STZ injection. At day 10 and month 5, the knockdown efficiency of Itgb1 was determined in the mice heart.

### Isolation of the nuclei from heart tissue

Isolation of nuclei from heart tissue were analyzed, as previously described (*McLellan et al., 2020*). In Brief, mouse heart tissues were homogenized using a Wheaton Dounce Tissue Grinder. Then, 3 ml of the homogenization buffer was added and incubated the homogenized tissue on ice for 5 min. Then, the homogenized tissue was filtered through a 40 mm cell strainer, mixed with an equal volume of working solution and loaded into the top of an OptiPrep density gradient on top of 5 ml of 35% Opti-Prep solution. The nuclei were separated through ultracentrifugation using an SW32 rotor (20 min, 9000 rpm). Thereafter, 3 ml of the nuclei were collected at the 29%/35% interphase and washed with 30 ml of phosphate-buffered saline (PBS) containing 0.04% bovine serum albumin (BSA). The nuclei were centrifuged at $300 \times g$ for 3 min and washed with 20 ml of PBS containing 0.04% BSA. Then, the nuclei were centrifuged at $300 \times g$ for 3 min and resuspended in 500 ml of PBS containing 0.04% BSA. All procedures were conducted on ice or at 4°C.

### Single-nucleus transcriptomic library preparation

Single-nucleus transcriptomic library preparation were performed as previously described (*Li et al., 2021*). Briefly, single nucleus was resuspended in PBS containing 0.04% BSA and added to each channel. The captured nucleus was lysed, and the RNA released was barcoded through reverse transcription in individual GEMs. Barcoded cDNA was amplified, and the quality was controlled using an Agilent 4200 TapeStation System. scRNA-seq libraries were prepared using Single Cell 3′ Library and Gel Bead Kit V3 following the manufacturer's introduction. Sequencing was performed on an Illumina Novaseq 6000 sequencer using a pair-end 150 bp (PE150) reading strategy (performed by Gene Denovo Biotechnology Co, Guangzhou, China).

### Clustering analysis

Alignment, filtering, barcode counting, and UMI counting were performed with Cell Ranger to generate a feature-barcode matrix and their global gene expression. Dimensionality reduction, visualization, and analysis of the scRNA-sequencing data were performed with the R package Seurat (version 3.1.2). As a further quality-control measure, nuclei that met any of the following criteria were filtered out: <500 or>4000 unique genes expressed, >8000 UMIs, or >10% of reads mapped to the mitochondria. After removing unwanted nuclei from the dataset, 2000 highly variable genes were used for the downstream clustering analysis. Principal component analysis (PCA) was performed, and the number of the significant principal components was calculated using the built-in 'ElbowPlot' function.

### Differentially expressed genes analysis

The expression level of each gene in the target cluster was compared with other cells using the Wilcoxon rank-sum test. The significantly upregulated genes were identified using the following criteria: (1) at least 1.28-fold overexpression in the target cluster, (2) expression in more than 25% of the cells belonging to the target cluster, and (3) FDR is less than 0.05.

### Gene set variation analysis

To identify cellular processes and pathways that were enriched in the different clusters, GSVA was performed using the GSVA R package (*Hänzelmann et al., 2013*) version 1.26, based on the cluster-averaged log-transformed expression matrix.

### GO and KEGG pathway enrichment analysis of the differentially expressed genes

GO and KEGG pathway enrichment analysis identified significantly enriched cellular processes and pathways in differentially expressed genes comparing with the whole genome background. The

calculated p-value was gone through FDR correction, taking FDR ≤0.05 as a threshold. GO term and KEGG pathways that met this condition were defined as the significantly enriched pathways in differentially expressed genes.

### Regulon analysis

Regulon analysis was performed on the SCENIC R package to carry out transcription factor network inference (*Aibar et al., 2017*). In brief, gene expression matrix was used as the input, and the pipeline was implanted in three steps. First, the gene co-expression network was identified via GENIE3 (*Huynh-Thu et al., 2010*). Second, we pruned each module based on a regulatory motif near a transcription start site via RcisTarget. Third, we scored the activity of each regulon for each single cell via theArea Under Curve (AUC) scores using AUCell R package.

### Statistical analyses

The statistical analysis of the results was performed using GraphPad Prism 9.0 software. Unpaired *t*-test or one-way analysis of variance was used to calculate the differences in mean values. A p-value ≤0.05 was considered statistically significant. Other statistical analyses not described above were performed using the ggpubr package in R software (https://github.com/kassambara/ggpubr copy archived at *Li, 2023*).

## Acknowledgements

We thank Guangzhou Gene Denovo Biotechnology Co (Guangzhou, China) for the technical support for single-cell RNA sequencing of normal and T2DM mouse heart, and thank Professor Huiling Zhang's team of the College of Pharmaceutical Science in Soochow University for their friendly help with the experiments on diabetic myocardial fibrosis in mice. Thanks to the public research platform of Soochow University for their technical support of this experiment. We would also like to thank the native English speaker from CureEdit who improved the grammar and readability of our manuscript.

## Additional information

### Funding

| Funder | Grant reference number | Author |
| --- | --- | --- |
| Suzhou Science and Technology Development Plan | SKJY 2021038 | Shigang Qiao |
| Jiangsu Key Talent Youth Awards in Medicine | QNRC2016219 | Shigang Qiao |
| Gusu Health Youth Talent Awards | GSWS2019092 | Shigang Qiao |
| Gusu Health Talent Program | GSWS2021068 | Zhenhao Zhang |
| Suzhou New District Science and Technology Project | 2020Z007 | Zhenhao Zhang |
| General Program of Basic Science in Jiangsu Higher Education Institutions | 21KJB350017 | Shudi Yang |
| the Core Medical Science Subjects in Suzhou | SZXK202131 | Shudi Yang |

The funders had no role in study design, data collection, and interpretation, or the decision to submit the work for publication.

## Author contributions
Wei Li, Conceptualization, Data curation, Formal analysis, Investigation, Methodology, Writing – original draft, Writing – review and editing; Xinqi Lou, Yingjie Zha, Yinyin Qin, Jun Zha, Lei Hong, Zhanli Xie, Formal analysis, Investigation; Shudi Yang, Conceptualization, Formal analysis, Investigation; Chen Wang, Conceptualization; Jianzhong An, Conceptualization, Supervision, Provide guidance during manuscript writing; Zhenhao Zhang, Conceptualization, Funding acquisition, Writing – original draft, Writing – review and editing, Investigation, Formal analysis; Shigang Qiao, Conceptualization, Formal analysis, Supervision, Funding acquisition, Investigation, Writing – original draft, Writing – review and editing

## Author ORCIDs
Zhenhao Zhang ⓘ http://orcid.org/0000-0001-5659-6315

## Ethics
This study was approved by the Ethics Committee of Soochow University and Suzhou Science & Technology Town Hospital, Gusu School, Nanjing Medical University. Reference number: 2022-B23. All mice were treated in accordance with the National Institutes of Health's Guidelines for the Care and Use of Experimental Animals (NIH publications no. 80-23, revised 1996).

## Decision letter and Author response
Decision letter https://doi.org/10.7554/eLife.80479.sa1
Author response https://doi.org/10.7554/eLife.80479.sa2

---

# Additional files

## Supplementary files
• Supplementary file 1. The 25 transcriptionally distinct pre-clusters with highly consistent expression patterns across individual mice hearts.

• Supplementary file 2. Genes with significant transcriptomic changes in the cardiac populations.

• Supplementary file 3. The top 10 upregulated genes in the cardiac populations.

• Supplementary file 4. Unique differentially expressed genes (uni-DEGs) in the cardiac populations.

• Supplementary file 5. Significantly differentially expressed genes in specific cell populations relative to others in mice hearts.

• Supplementary file 6. Cell type-specific receptors in the cardiac populations.

• Supplementary file 7. Cell type-specific ligands in the cardiac populations.

• Supplementary file 8. Relative expression of a selection of essential growth factors across the major cardiac cell types.

• Supplementary file 9. The number of ligand–receptor pairs among the cardiac cell populations in the healthy mice.

• Supplementary file 10. Ligand–receptor pairs among the cardiac cell populations in the healthy mice.

• Supplementary file 11. Significantly differentially expressed ligands in each cell population.

• Supplementary file 12. Significantly differentially expressed receptors in each cell population.

• Supplementary file 13. The cognate ligands of Egfr.

• Supplementary file 14. The cognate ligands of Pdgfra.

• Supplementary file 15. Ten transcriptionally distinct fibroblast subpopulations.

• Supplementary file 16. Distinct signatures of each fibroblast subpopulations in mice hearts.

• Supplementary file 17. The transcription factor gene-regulatory networks in the distinct subpopulations.

• Supplementary file 18. Gene expression of the transcription factors shown in *Figure 5A, B*.

• Supplementary file 19. Genes with significant transcriptomic changes in each fibroblast subpopulation.

• Supplementary file 20. Unique differentially expressed genes (uni-DEGs) in each fibroblast subpopulation.

- Supplementary file 21. The cognate ligands of Itgb1.
- MDAR checklist

## Data availability

All sequencing data that support this study is available from the Genome Sequence Archive in BIG Data Center (http://bigd.big.ac.cn/) with the accession code CRA007245. The ligand and receptor pairing dataset was obtained from Fantom5 (https://fantom.gsc.riken.jp/5/suppl/Ramilowski_et_al_2015/), as described in *Ramilowski et al., 2015*. Source data files are provided to support CT values of Col1a1 and Col3a1 used for Figure 3G,H. Source data files are provided to support CT values of Itgb1 used for Figure 6H. Source data files are provided to support Table 1 in Figure 3—Figure supplement 5.

The following dataset was generated:

| Author(s) | Year | Dataset title | Dataset URL | Database and Identifier |
|---|---|---|---|---|
| Li W, Lou X, Zha Y, Zha J, Hong L, Xie Z, Yang S, Wang C, An J, Zhang Z, Qiao S | 2022 | Single-Cell RNA-seq of Heart Reveals Intercellular Communication Drivers of Myocardial Fibrosis in Diabetic Mice | https://ngdc.cncb.ac.cn/search/?dbId=gsa&q=CRA007245 | CNCB Genome Sequence Archive, CRA007245 |

The following previously published dataset was used:

| Author(s) | Year | Dataset title | Dataset URL | Database and Identifier |
|---|---|---|---|---|
| Ramilowski JA, Goldberg T, Harshbarger J, Kloppman E, Lizio M, Satagopam VP, Itoh M, Kawaji H, Carninci P, Rost B, Forrest ARR | 2015 | A draft network of ligand-receptor-mediated multicellular signalling in human | https://fantom.gsc.riken.jp/5/suppl/Ramilowski_et_al_2015/ | Fantom5, fantom |

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
