## [Editor Report]

In this work, utilizing a murine model as well as in vitro studies, the authors provide novel insights about drivers of intercellular communication underlying pathological extracellular matrix remodeling during diabetic cardiomyopathy. The results of the study will be of interest to investigators interested in diabetic cardiomyopathy and heart failure as well as those looking for new potential targets in the treatment of heart failure.

---

## [Decision Letter]

**Decision letter after peer review:**

Thank you for submitting your article "Single-Cell RNA-seq of Heart Reveals Intercellular Communication Drivers of Myocardial Fibrosis in Diabetic Mice" for consideration by *eLife*. Your article has been reviewed by 3 peer reviewers, including Muthuswamy Balasubramanyam as Reviewing Editor and Reviewer #1, and the evaluation has been overseen by a Reviewing Editor and a Senior Editor. The following individual involved in the review of your submission has agreed to reveal their identity: Paolo Madeddu (Reviewer #3).

The reviewers have discussed their reviews with one another, and the Reviewing Editor has drafted this letter to help you prepare a revised submission.

Essential revisions:

*Reviewer #1 (Recommendations for the authors):*

The manuscript does have some limitations which need to be addressed via the review process. The study in its technical components and presentation are appropriate.

Comments:

1. Overall, the manuscript needs some improvement in results and discussion sessions. Authors are advised to discuss their results in consonance with the recent findings illustrated in the following recent important articles pertinent to the manuscript topic.

Koenig A.L. et al. 2022 (Nature Cardiovascular Research)

Ackers-Johnson et al. 2018 (Nature Communications)

2. Legends to the figures need to be shortened – as there is an overlap in the figure legends and the methodology section.

3. In the introduction, lines 78-88 should be deleted as the text context belongs to the Results section.

*Reviewer #2 (Recommendations for the authors):*

1) Figure2 S1: Please highlight some examples of what is displayed.

2) 2C and D: Please use the same color code for the panels.

3) 2F, G please label the y-axis.

*Reviewer #3 (Recommendations for the authors):*

The study is technically sound and well-illustrated. Some sentences need to be revisited. For instance, in the Introduction "Since adult mammalian cardiomyocytes are virtually incapable of regeneration, the most extensive extracellular matrix remodeling and fibrosis of the heart occurs in diseases caused by acute cardiomyocyte death". Diabetes does not cause acute cardiomyocyte death.

In other points, associative findings are used to state a cause-relation effect or to highlight obvious concepts such as fibroblasts are important in fibrosis.

The study will take advantage of confirming the main data with the use of an additional mouse model. In addition, female mice should be included. The most important limit is represented by the fact the authors studied at a very late stage, these changes may not be the ones responsible for the final phenotype. A pseudotime analysis adding an early analysis of gene expression at a single cell level is mandatory.

Functional data are missing and histology is limited to fibrosis. What about capillary density and cardiomyocytes? Any effect on these cells by the used treatments? What about diastolic dysfunction? To understand whether the improvements are significant, an analysis by echocardiography is relevant.

The human data are not very useful. The association of diabetes with variants does not add to the experimental evidence provided on fibrosis. This part is weak and may be eliminated unless they provide evidence of an association with cardiac fibrosis or diastolic dysfunction.

General Recommendations for the authors:

The manuscript should be checked and corrected by a colleague whose first language is English. Also, the manuscript title should be changed from "Single-Cell RNA-seq of Heart Reveals Intercellular Communication Drivers of Myocardial Fibrosis in Diabetic Mice" to "Single-Cell RNA-seq of Cardiomyocytes Reveals Intercellular Communication Drivers of Myocardial Fibrosis in Diabetic Cardiomyopathy".

---

## [Author Response]

Reviewer #1 (Recommendations for the authors):The manuscript does have some limitations which need to be addressed via the review process. The study in its technical components and presentation are appropriate.Comments:1. Overall, the manuscript needs some improvement in results and discussion sessions. Authors are advised to discuss their results in consonance with the recent findings illustrated in the following recent important articles pertinent to the manuscript topic.Koenig A.L. et al. 2022 (Nature Cardiovascular Research)Ackers-Johnson et al. 2018 (Nature Communications)

Thank you very much for the comments of the reviewing editor. We have carefully read the paper recommended by the editor and have set forth these important revisions in the results and discussions section of the revised manuscript.

2. Legends to the figures need to be shortened – as there is an overlap in the figure legends and the methodology section.

Thank you very much for the comments and suggestions of the editor. We have carefully revised the figure legends in the main text and updated the manuscript.

3. In the introduction, lines 78-88 should be deleted as the text context belongs to the Results section.

Thanks to the editor for the precious comments. We have deleted the lines 78-88 of the introduction in the revised manuscript.

Reviewer #2 (Recommendations for the authors):1) Figure2 S1: Please highlight some examples of what is displayed.

Thanks to the reviewer for these comments. Details are as follow: Figure 2 is intended to give the reader a macroscopic view of the receptor and ligand expression patterns of different cell populations. More specific expression information can be seen in Supplementary file 6 and 7.

2) 2C and D: Please use the same color code for the panels.

Thank you very much for the opinion of the reviewer. We have remade Figure

2C and D according to your suggestion in the revised manuscript.

3) 2F, G please label the y-axis.

Thank you very much for the comment of the reviewer. We have remade Figure 2F and G according to your suggestion in the revised manuscript.

Reviewer #3 (Recommendations for the authors):The study is technically sound and well-illustrated. Some sentences need to be revisited. For instance, in the Introduction "Since adult mammalian cardiomyocytes are virtually incapable of regeneration, the most extensive extracellular matrix remodeling and fibrosis of the heart occurs in diseases caused by acute cardiomyocyte death". Diabetes does not cause acute cardiomyocyte death.

Thank you very much for the comments of the reviewer. We have carefully read the introduction and revised the description according to your comments. Please refer to the newly submitted revised manuscript for specific revision.

In other points, associative findings are used to state a cause-relation effect or to highlight obvious concepts such as fibroblasts are important in fibrosis.The study will take advantage of confirming the main data with the use of an additional mouse model. In addition, female mice should be included. The most important limit is represented by the fact the authors studied at a very late stage, these changes may not be the ones responsible for the final phenotype. A pseudotime analysis adding an early analysis of gene expression at a single cell level is mandatory.Functional data are missing and histology is limited to fibrosis. What about capillary density and cardiomyocytes? Any effect on these cells by the used treatments? What about diastolic dysfunction? To understand whether the improvements are significant, an analysis by echocardiography is relevant.

Thanks to the reviewer for these precious comments. At your request, we have reinserted the DB/DB transgenic diabetic mice model and the female C57 mice in combination with high-fat-diet (HFD)/streptozotocin (STZ)-induced diabetes model. We found that the Pdgf(s)-Pdgfra axis remained activated in DB/DB and female C57 diabetic mice, and that the Hrc+ fibroblast subpopulation plays an important role in myocardial fibrosis. Finally, Ima treatment significantly inhibited Pdgfra

Phosphorylation, Pdgf(s)-Pdgfra interaction and collagen accumulation in transgenic DB/DB (Figure 7) and female C57 diabetic mice (Figure 8). Please see the revised manuscript for details.

In order to understand the improvement of heart function in diabetic mice after Ima treatment, we assessed the heart function of C57 diabetic mice by echocardiography, and the results showed that the heart function of mice was significantly improved after Ima treatment. Please check the specific results (Figure 3—figure supplement 5) in the main text of newly revised manuscript. At the same time, we also assessed myocardial hypertrophy by WGA staining (Figure 3—figure supplement 4), and the results showed that Ima treatment significantly inhibited myocardial hypertrophy in diabetic mice. The above validation experiments have fully revealed that Ima treatment can significantly inhibit myocardial fibrosis and improve cardiac function in diabetic mice. Therefore, Pdgf(s)-Pdgfra interaction and appearance of Hrc+ activated fibroblasts could be a common mechanism that promotes cardiac fibrosis in type II diabetes.

Finally, to investigate the transition of the fibroblast clusters in the mice heart, we applied single-cell trajectories analysis and showed that the cells formed a continuous progression, then progressively ending toward Hrc^hi^ fibroblast, indicating that Hrc^hi^ fibroblast may be a myocardial fibrosis-related cardiac fibroblast subpopulation (Author response image 1). Thanks again to the reviewer for these professional and precious comments.

**Author response image 1. sa2fig1:** Single-cell trajectories analysis.

The human data are not very useful. The association of diabetes with variants does not add to the experimental evidence provided on fibrosis. This part is weak and may be eliminated unless they provide evidence of an association with cardiac fibrosis or diastolic dysfunction.

Thank you very much for the opinion of the reviewer. We have deleted the results of the association of Itgb1 variants with cardiac fibrosis. Thanks again to the reviewer for these professional comments.

General Recommendations for the authors:The manuscript should be checked and corrected by a colleague whose first language is English. Also, the manuscript title should be changed from "Single-Cell RNA-seq of Heart Reveals Intercellular Communication Drivers of Myocardial Fibrosis in Diabetic Mice" to "Single-Cell RNA-seq of Cardiomyocytes Reveals Intercellular Communication Drivers of Myocardial Fibrosis in Diabetic Cardiomyopathy".

Thanks to the reviewer for these kind comments. The revised manuscript has been checked by CureEdit to further improve the grammar and readability. At the same time, we have changed the title of the manuscript to "Single-Cell RNA-seq of Heart Reveals Intercellular Communication Drivers of Myocardial Fibrosis in Diabetic Cardiomyopathy" based on your comments. Thanks again to the reviewer for these precious and professional comments.